# Adapting to altered auditory cues: Generalization from manual reaching to head pointing

Chiara Valzolgher[1,2]*, Michela Todeschini[3], Gregoire Verdelet[1,4], Julie Gatel[5], Romeo Salemme[1,4], Valerie Gaveau[1,6], Eric Truy[5], Alessandro Farnè[1,2,4], Francesco Pavani[1,2]

1 Integrative, Multisensory, Perception, Action and Cognition Team (IMPACT), Lyon Neuroscience Research Center, Lyon, France, 2 Center for Mind/Brain Sciences—CIMeC, University of Trento, Trento, Italy, 3 Department of Psychology and Cognitive Sciences (DiPSCo), University of Trento, Trento, Italy, 4 Neuroimmersion, Lyon Neuroscience Research Center, Lyon, France, 5 Hospices Civils de Lyon, Lyon, France, 6 University of Lyon 1, Villeurbanne, France

* chiara.valzolgher@unitn.it

**Data Availability Statement:** Data and R codes can be retrieved from osf.io/dt76y.

**Funding:** The study was supported by a grant of the Agence Nationale de la Recherche (ANR-16-

## Abstract

Localising sounds means having the ability to process auditory cues deriving from the interplay among sound waves, the head and the ears. When auditory cues change because of temporary or permanent hearing loss, sound localization becomes difficult and uncertain. The brain can adapt to altered auditory cues throughout life and multisensory training can promote the relearning of spatial hearing skills. Here, we study the training potentials of sound-oriented motor behaviour to test if a training based on manual actions toward sounds can learning effects that generalize to different auditory spatial tasks. We assessed spatial hearing relearning in normal hearing adults with a plugged ear by using visual virtual reality and body motion tracking. Participants performed two auditory tasks that entail explicit and implicit processing of sound position (head-pointing sound localization and audio-visual attention cueing, respectively), before and after having received a spatial training session in which they identified sound position by reaching to auditory sources nearby. Using a cross-over design, the effects of the above-mentioned spatial training were compared to a control condition involving the same physical stimuli, but different task demands (i.e., a non-spatial discrimination of amplitude modulations in the sound). According to our findings, spatial hearing in one-ear plugged participants improved more after reaching to sound trainings rather than in the control condition. Training by reaching also modified head-movement behaviour during listening. Crucially, the improvements observed during training generalize also to a different sound localization task, possibly as a consequence of acquired and novel head-movement strategies.

## Introduction

Humans and other hearing species can localize sounds in space. Spatial hearing relies on the interpretation of binaural auditory cues, resulting from the different inputs reaching the two

CE17-0016, VIRTUALHEARING3D, France, https://anr.fr/) to F.P. and A.F. In addition, C.V. was supported by a grant of the Università Italo-Francese/Université Franco-Italienne (https://www.universite-franco-italienne.org/), the Zegna Founder's Scholarship (https://www.zegnagroup.com/it/csr/founder-scholarship/) and Associazione Amici di Claudio Demattè (http://www.amicidematte.org/). F.P. and A.F. were also supported by a prize of the Foundation Medisite (France), by the Neurodis Foundation (France) and by a grant from the Italian Ministry for Research and University (MUR, PRIN 20177894ZH) (https://www.miur.gov.it/). Finally, A.F. was supported by the IHU CaSaMe ANR-10-UBHU-0003 and ANR 2019CE37 Blind Touch (ANR: https://anr.fr/). The funders had no role in study design, data collection and analysis, decision to publish, or preparation of the manuscript.

**Competing interests:** The authors have declared that no competing interests exist.

ears, and monaural cues, resulting from the amplitude and spectral changes occurring in the single ear [1–3]. Although listeners experience their spatial hearing skills as constant and stable, temporary or permanent conditions can alter the auditory cues and affect this fundamental hearing ability. Examples include partial or complete hearing loss to one ear [4], age-related hearing loss [5], use of hearing aids [6] or use of cochlear implants [7]. Yet, in the last decades research has shown that the relearning of spatial hearing skills is possible, in humans [8–13] as well as in other animals [14].

Adaptation to new auditory cues can be achieved using multisensory cues to sound position (for reviews see: [8–10], 15, 16]). Training with audio-visual stimuli can be more effective than training with auditory information alone [11, 12, 17, 18]. In addition, adapting to new auditory cues may be easier through motor interactions with the sound sources, using tasks in which acoustic stimulation results from the subjects' own movements [13] or in which participants are encouraged to act towards sounds [19, 20]. For instance, participants could be asked to hit a moving sound presented in a virtual auditory space using a hand-held tool [21, 22], or to shoot audio-visual moving targets in virtual reality [23]. Other relevant examples include the work of Steadman and colleagues, in which participants were required to actively point their head toward sounds [24]. Taken together, these works suggest that motor interactions with sounds could promote adaptation to new auditory cues.

In a recent study [25], the above-mentioned issue was addressed by asking to one-ear-plugged normal hearing participants to identify the position of sounds by reaching or by naming the labels associated with each speaker. The manipulation was performed across groups, using virtual reality (VR). Seventeen virtual speakers were presented to participants in a virtual room; participants' hand movements and head rotations were monitored through kinematic tracking. Both groups received audio-visual feedback about their performance on a trial-by-trial basis. Importantly, participants in both groups were always allowed to move their head during listening, with sounds lasting until response (approximatively 4 seconds). Results show a faster reduction in the number of errors in the reaching group than in the naming one. This suggests that reaching to sounds plays a specific role when listeners must adapt to new auditory cues. Moreover, the reaching group increased head exploration movements during listening, and these head-movements led to sound localization improvements. Specifically, the improvements determined by reaching to sounds were related to changes in the amount of space explored with the head. This suggests a potential role of head movements during listening in this adaptive behavior.

In this recent work [25] we also documented that reaching to sounds can ameliorate sound localization with one ear plugged on a trial-by-trial basis. However, it remains unclear if participants improved because of practice with the specific auditory task or if they learned new and effective ways to adapt to the altered auditory cues instead. If the latter answer is true, then sensorimotor training should improve sound source localization also when the task entails different sound source positions, and when it requires a response with a non-trained effector (i.e. the head rather than the hand). In the present study, this hypothesis was addressed directly by testing if performance improvement induced by reaching to sounds can extend (i.e., generalize) beyond the trained auditory task itself.

To this aim, we recruited a group of normal hearing participants and we temporarily degraded their spatial hearing by plugging one ear. We used the reaching task developed in our previous work [25] to train their sound localization. Crucially, before and after this training participants were also tested in two tasks aiming at revealing generalization effects: a head-pointing sound localization task [25] and an audio-visual attention cueing task [26]. The head-pointing localization task required to explicitly localize sounds and differed from training in terms of visual scenario (speaker position no longer visible), spatial position of the targets

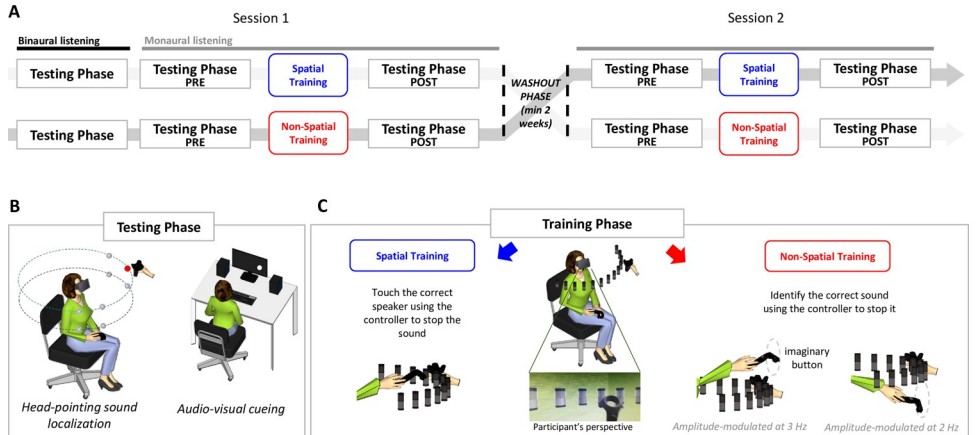

**Fig 1. Experimental procedure and setting.** (A) Experimental crossover protocol: all participants took part in 2 training session (session 1 and session 2). Each session was composed by two Testing phases (termed, 'pre' and 'post'), separated by a Training phase (Spatial or Non-Spatial, across sessions). All phases were performed in monaural listening (i.e. with one ear plugged), but an additional testing phase at the beginning of session 1 measured performance in binaural listening. (B) Testing Phases. To the left, a schematic representation of participant wearing the HMD and holding the controller used for validation during head-pointing sound localization. The eight spheres in front of the participant indicate the pre-determined speaker positions, the loudspeaker held by the experimenter's hand illustrates stimulation position in one example trial. To the right, a schematic representation of the setting for the audio-visual attention cueing task. (C) Training Phase. In the center, a schematic representation of participant wearing the HMD and holding the controller. Thirteen cylindrical visible speakers indicate the pre-determined positions during each of the training sessions. Below, a close-up of the scene as visible inside the HMD from participant's perspective: during the task participant saw the virtual room, the visible speakers and the controller they held in their hand. Cartoons on either side illustrate example of the response's movements performed during the Spatial (left) and Non-Spatial (right) training.

(different azimuths and two different elevation of sound position) and from response demands (pointing with the head instead of reaching with the hand). The rationale for these changes was to minimize potential carry-over of mere sensorimotor adaptations acquired during the spatial training to the novel auditory task. The audio-visual attention cueing task was instead an implicit sound localization task: unlike head-pointing sound localization participants were never asked to explicitly indicate sound position and sounds only served as lateralized attention-orienting cues for the discrimination of visual targets (i.e., an audio-visual analogue to classic attention cueing paradigms) [26–29]. This second test was introduced to probe whether adaptation to new auditory cues could also impact audio-visual attention orienting, a skill that can be hampered by monaural listening [26].

Using a crossover experimental design (Fig 1), a training in which participants performed reaching movement to identify sound position (hereafter referred to as Spatial training) was compared to a control training in which participants performed comparable reaching movements to identify a non-spatial feature of the sound (from now on, the Non-Spatial training). During the Non-Spatial training participants were asked to discriminate between sounds with two different amplitude modulations rather than to focus on the spatial position of the sources. Each participant was tested in both training conditions in two successive sessions, separated by a wash-out period of 2-weeks minimum. While the two trainings used identical physical stimuli, the behavioral requests differed and entailed the processing of spatial vs. non-spatial features of sounds. These different task demands recruit substantially different cognitive and brain mechanisms, as revealed by converging evidence from neurophysiology [30], neuropsychology [31] and cognitive neuroscience [32–34]. Our key prediction was to observe generalization of training effects after the Spatial training more than after the Non-Spatial training paradigm.

## Methods

### Participants

Twenty participants (age: M = 29.4, SD = 10.5, 5 males, 19 right-handed) were recruited to participate in the study, carried out in the otolaryngology department of the civil Hospital Edouard Herriot (HEH) in Lyon (France). Previous findings from our laboratory [25] showed that sound localization improvements emerge following a training similar to the one proposed here (see 'Spatial Training' below), with an effect size ($\eta_2$) of 0.24 (which corresponds to Cohen's d of 1.12) [35]. Using the G*Power, we calculated that to obtain a similar effect (alpha = 0.05, Power = 0.99) the sample size required was at least 17 participants. Thus, we decided to include 20 participants.

All participants signed an informed consent before starting the experiment, which certified the ethical approval of the national ethics committee in France (Ile de France X, N˚ ID RCB 2019-A02293-54), and recorded in clinicaltrials.gov (NCT04183348). All participants had normal to corrected-to-normal vision and reported no motor or vestibular deficit as well as no history of neurological or psychiatric disorders. Hearing thresholds were measured for all participants using an audiometer (Equinox 2.0, Interacoustics), testing different frequencies (250, 500, 1000, 2000, 4000, 8000 Hz), on the right and left ear separately. All participants had an average threshold below 20 dB HL for both ears.

### General structure of the experimental session

Participants were invited to participate in two experimental sessions, separated by at least 2 weeks. Each session comprised three phases: two testing phases and one training (Fig 1A). The two experimental sessions differed exclusively in the training phase task: firstly, participants were involved in the experimental training task (Spatial Training), and later in the control training task (Non-Spatial Training). In this way, all participants performed both training types in a within-subject design. Order of training type was counterbalanced across participants. The testing phases were identical in both experimental sessions: they included the head-pointing sound localization (conducted in VR) and the audio-visual attention cueing task (conducted outside VR).

Participants completed the entire experiment in monaural listening. This temporary auditory cue alteration was obtained by occluding the right ear of the participant with a plug (3M PP 01 002; attenuation values: high frequencies = 30 dB SPL; medium frequencies = 24 dB SPL; low frequencies = 22 dB SPL) and a monaural ear muffs (3M 1445 modified to cover only the right ear; attenuation values: high frequencies = 32 dB SPL; medium frequencies = 29 dB SPL; low frequencies = 23 dB SPL). At the beginning of their first session, participants also completed the testing phase without the ear plug (Fig 1A). This provided a baseline measure of their spatial hearing skills in binaural listening before exposure to monaural listening.

### Apparatus

Virtual reality and kinematic tracking were implemented using the HTC Vive (Vive Enterprise). The system comprised one head-mounted display (HMD, resolution: 1080 x 1200 px, Field Of View (FOV): 110˚, Refresh rate: 90 Hz), 2 hand-held controllers (one held by the experimenter to calibrate head center, and one held by participants to interact with the virtual environment during the training phase), 1 tracker mounted above the speaker to track its position in real time, and 2 lighthouse base-stations (Lighthouse V1.0) for scanning the position of the controller, trackers and the HMD. Tracking precision and accuracy of the HTC Vive System are adequate for behavioral research purposes [36]. Stimuli were controlled and delivered

using a LDLC ZALMAN PC (OS: Windows 10 (10.0.0) 64bit; Graphic card: NVIDIA GeForce GTX 1060 6GB; Processor: Intel Core i7-7700K, Quad-Core 4.2 GHz/4.5 GHz Turbo—Cache 8 Mo—TDP 95W) using Steam VR software and the development platform Unity3D (Unity Technologies, San Francisco, CA).

Real free-field auditory stimuli were delivered by a loudspeaker (JBL GO Portable from Harman International Industries, Northridge, California USA, 68.3 x 82.7 x 30.8 mm, Output Power 3.0W, Frequency response 180 Hz– 20 kHz, Signal-to-noise ration > 80 dB), with the HTC Vive tracker firmly attached to its top. During the entire experiment we tracked the loud-speaker position, as well as the position of the controller held in the participant's hand and the HMD, via a dead reckoning process using gyroscope and accelerometer, plus a correction sig-nal from the lighthouse system every 8.333 milliseconds. Both tracking method allowed posi-tion tracking with a frequency sample of 250 Hz. The software is designed to guide the experimenter to align the real loudspeaker (i.e., the sound source) with a set of pre-determined positions defined in the 3D virtual environment in each trial. This method combining virtual reality and kinematic tracking to measure sound localization abilities has been developed in our laboratory [37] and has been already adopted in previous studies [25, 38].

The audio-visual attention cueing test was carried out without VR, with the participant sit-ting at a desk. The apparatus for this task included a separate PC, a DELL 24" monitor, a key-board and two speakers, positioned at ear level on both sides of the screen, each located 20˚ to the left or to the right of central fixation (see Fig 1B). The height of the chair on which the par-ticipants sat was adjusted to favour the support of the head on the chin rest on the edge of the table, aligned with the centre of the monitor. The test consists of a visual discrimination task implemented with the program OpenSesame®.

## Procedure and stimuli

Before starting the experiment, participants were informed about the use of the VR equipment. When engaged in the VR tasks participants sat on a rotating armless chair with no chin rest, placed in the center of the room. The room (3.6 m x 3.9 m, height 2.7 m) was quiet and treated with sound-proof panels to partially reduce echoes. The T60 reverberation of the room was 0,30–0,33 seconds, as measured using a Blue Solo 01bB phonometer. Instruction for each task were provided immediately before the task started.

**Testing phase.**    *Head-pointing sound localization.* This part of the experiment was carried out entirely in VR. The participant was immersed in a virtual room with green walls, reproduc-ing exactly the size and shape of the real room. The room was devoid of any objects, except for light source on the ceiling and a wooden door behind the subject. The task comprised 40 actual trials, plus 5 practice trials presented at the beginning of the task. At the beginning of each trial, participants were asked to direct their gaze in front of them to align their head with a white central fixation cross. As soon as the head correct position was reached, the fixation cross turned blue. This procedure ensured that initial posture was comparable across trials and participants, even in the absence of a chin-rest. In the meanwhile, the experimenter placed the speaker at one of the possible eight pre-determined positions. Eight predetermined positions were used throughout the experiment, resulting from the combination of 4 different azimuths in the frontal space in respect to participant's head position (-67.5˚, -22.5˚, 22.5˚ or 67.5˚), 2 different elevations (+5 and -15˚) and a single distance (55 cm) (Fig 1A). Each position was reached manually by the experimenter using visual indications provided on the dedicated instruction monitor. When the correct starting head posture was reached, and the loudspeaker was positioned correctly (i.e., within a sphere with diameter of 5 cm centred on the pre-deter-mined location), the target sound was delivered.

The target sound consisted of 3 second white noise bursts, amplitude-modulated at 2.5 Hz, and delivered at about 65 dB SPL, as measured from the participant's head. During target sound delivery the fixation cross turned white. From that moment, participants were free to move their heads and rotate the chair they sat on to explore the surrounding space. The task consisted in localizing the exact source of the sound and to indicate it using the head direction as a pointer. This response was aided by the fact that the visible fixation cross in the HMD followed head direction displacements. It is noteworthy that, since the speaker was invisible in VR, participants did not have visual cues about sound source position. At the end of the 3 seconds of sound presentation, the central cross turned red, to indicate to the participants to validate their response (i.e., their current head direction) by pressing the button at the base of their hand-held controller.

Participants were informed that sounds could be delivered anywhere in the 3D around them and they did not have to judge the distance but only the elevation and the azimuth dimension of the sound space. Note that head and trunk movements remained unconstrained both during and after sound emission, allowing spontaneous active listening behaviour (e.g., orienting the head in the direction of the sound even during the sound emission). This task lasted approximately 10 minutes.

*Audio-visual attention cueing task.* This part of the experiment was carried out entirely outside VR. Participants sat at the experimental table, placed inside the same room in which the VR experiments are carried out. Unlike all the VR tasks, in this audio-visual cueing task participants rested their heads on the chinrest, hence no head-movements were ever allowed.

Each trial started with a white fixation cross appearing in the centre of the monitor and remaining visible until response. After a random delay (450–600 ms), an auditory stimulus (white noise, duration 100 ms) was emitted from one of the two loudspeakers. Loudness of the auditory stimulus was approximately 60 dB SPL, as measured from head position. After 100 ms from sound delivery, the visual target was presented. This consisted of a white filled circle (20 pixels radius, 0.5˚ of visual angle), appearing on a black background for 140 ms in the upper or lower hemifield with respect to the horizontal meridian passing through visual fixation (1.15˚ above or below the meridian), either in the left or right hemifield (10˚ from fixation). In half of the trials, the visual target and the sound appeared on the same hemispace (congruent trials), whereas on the remaining half the visual target and the sound appeared on opposite hemispaces (incongruent trials).

Participants were asked to keep their gaze toward central fixation throughout the task and to indicate as quickly and accurately as possible the elevation of visual targets appearing one at a time on either side of fixation. Up/down responses were given using the up/down arrows keys on an external keyboard, using the index-middle fingers of the right hand. Participants had a timeout of 2000 ms to give their answer. The experiment started with 8 practice trials, following by 128 trials divided in 2 blocks randomly divided in congruent and incongruent audio-visual conditions and it lasted 10 minutes approximatively (Pavani et al., 2017). Participants received feedback on accuracy (percentage of correct responses) and mean response time (in ms) only at the end of each block. Importantly, they were also explicitly informed that sounds were always entirely task-irrelevant.

**Training phase.** This part of the experiment was carried out entirely in VR. In each session, this phase was either a Spatial or a Non-Spatial training (see below). Both training tasks took place in the same virtual room used for the head-pointing sound localization task. The only difference was that during both training tasks, thirteen virtual speakers were visible in front of participants. They were arranged in a semicircle distributed in front of participants spanning between ± 72˚ (12˚ between each of them). The distance to the participant was always 55 cm (Fig 1C). All sounds were presented just below ear level (-5˚ offset between

tracker and speaker center). Note that sound positions were different from head-pointing sound localization.

Target sounds were delivered from the same azimuth and elevation as the virtual speakers, with a small distance offset so that the real speaker was 5 cm further away from the virtual speaker, to avoid collisions between the controller and the real speaker during the reaching response (see below). The target stimulus always consisted of a white noise: half of the stimuli were amplitude-modulated at 2 Hz and the remaining half at 3 Hz, to create clearly distinguishable targets. Targets sounds were delivered from each of the virtual speaker in random order (12 repetition for each of the 13 speakers, resulting in 156 trials overall, divided into 3 blocks of 52 trials each).

Crucially, exactly the same stimuli were delivered in the two training tasks, thus making the auditory component of the two trainings identical in all respects. In addition, they both involved a similar motor response: a reaching movement. Participants were informed that at the beginning of each trial the controller had to be held in the right hand, at sternum level, and that once the response was completed it had to return to that starting position. However, for the Spatial training the reaching movements served to indicate the perceived sound position, whereas for the Non-Spatial training they served to indicate the perceived amplitude modulation in the target sound.

*Spatial training*. Participants had to identify the speaker from which the sound was coming by reaching it using the controller in the right hand (Fig 1C). To prevent participants from colliding with the speaker held in the experimenter's hand, the setting provides that the real speakers was moved back 5 centimeters in depth, which does not affect the directionality of the perceived sound. Participants received a brief vibration of the controller upon contact with the chosen speaker, irrespective of whether the response was correct or incorrect. Importantly, if they reached and touched the correct speaker, the sound stopped. On the contrary, if participants reached the wrong speaker they received a visual feedback: the correct speaker started to flash. Specifically, from the correct source location, a series of red concentric circles (1024x1024 px, 2 circles per second) expanded intermittently, irradiating the surrounding space. The rationale was to capture participants attention even if they were looking toward a different zone of the space, including the opposite hemispace. The visual feedback and the sound stopped only when the subject reached the correct speaker position with the controller. This has two implications: first, a sense of agency was associated with the correct response; second, whenever the wrong speaker was originally selected, a combination of visual and auditory signals guided the participant to the correct sound source. The entire training lasted about 25 minutes.

*Non-Spatial training*. Participants had to identify whether the emitted sound amplitude was modulated at faster (3 Hz) or slower (2 Hz) rate (note that participants accustomed with the two auditory stimuli before starting the training). For fast amplitude-modulated sounds, participants directed the controller in front of them, above the row of speakers arranged at head height, aiming to touch an invisible virtual button above the central speaker. For fast amplitude-modulated sounds, participants had to reach an invisible virtual button below the same central speaker (Fig 1C). A vibration from the controller indicated that one of two buttons was correctly reached. As in the Spatial training feedback procedure, a visual feedback was delivered in case of erroneous responses. This was a series of red concentric circles that expanded intermittently from above or below the central speaker to indicate the correct position to reach. Recall that target sounds were nonetheless presented from different spatial positions during the training, although this spatial information was totally task-irrelevant. The entire training lasted about 15 minutes.

## Analyses

Statistical analyses were run using R (version 1.0.143) (R Core Team, 2013). For the linear mixed-effect (LME) model, we used the R-packages emmean, lme4, lmerTest in R Studio [39, 40]. The R-package car was used to obtain deviance tables from the LME models. When appropriate, we calculated Cohen's $d_{av}$ as index of effect size [41].

**Performance.** *Head-pointing sound localization*. To study performance in the head-pointing sound localization task, we focused on absolute error (i.e., the absolute deviation of the mean response from the source position) and signed error (i.e., the signed difference between the source and the response) in azimuth and elevation, separately. Signed error was negative or positive, to indicate an overall bias to respond. All variables were calculated for each individual trial. Trials in which the participant had problems with response validation (e.g., they responded before the end of sound emission), or in which the HMD signal were lost, were removed from the analyses (14 trials out of 4000 have been removed).

*Audio-Visual Cueing task*. To study performance in the Audio-Visual Cueing task we calculated the Cueing Effect (CE), expressed in milliseconds. The CE was calculated by subtracting the average reaction time (RTs) when the auditory cue and the visual target occur on the same side of space, from that obtained when the auditory cue and the visual target occur on opposite sides of space (27). The calculation was performed separately for each participant, phase and training. Trials with incorrect responses (i.e., wrong elevation judgement on the visual target) were excluded from the analyses (369 trials out of 12800).

*Training task*. To study changes in sound localization performance during the Spatial Training, we examined the absolute error (degrees) and the signed error (degrees) during the reaching to sound task. As target position only changed in the horizontal dimension, all errors were computed only in azimuth. The percentage of correct answers was considered to study the performance during the Non-Spatial training.

**Spontaneous head-movements.** Head movements were measured in all VR tasks (i.e., head-pointing to sounds, reaching to sounds during the Spatial Training, reaching to indicate amplitude-modulation differences in the Non-Spatial training).

The tangential velocity of the head on the x, y, z axis (expressed in degrees of rotation) using two-points central difference derivate algorithm [42] with 5 points for the half-window (smoothing purpose) was calculated to study head movements. The onset and the end of the movements were computed automatically using a velocity-based threshold (10°/s) [25]. Each head-movement was checked manually by visualizing the spatial rotation changes of the head and its speed using a custom-made tool box in MATLAB R2018b. The rationale was to eliminate trials in case the HMD data were lost.

We focused the analysis on three dependent variables: the number of head-movements, the head-rotation extent around the vertical axis and the head-rotation bias (i.e., center of gravity of head-rotation). To compute the number of head-movements, all the detected peaks of velocity in the head trace were considered, yet all movements smaller than 2° degrees to avoid noise were removed (i.e. excluding movements which are not indicators of spontaneous head intentional movements and may reflect micro postural movements not related to the task). To calculate head-rotation extent we sum the absolute value of the rightward and leftward head-rotation extremity. For instance, if the head rotated 20° to the right and 40° to the left, the head-rotation extent was calculated as the sum of the two, hence 60°. To calculate head-rotation bias, we sum values of the rightward and leftward head-rotation extremity. For instance, if the head rotated 20° to the right and 40° to the left (left is expressed with negative sign: -40°), the head-rotation bias was calculated as the signed sum of the two, therefore -20°.

Data and R codes can be retrieved from osf.io/dt76y. This study was not preregistered.

## Results

### Does monaural listening increase sound localization errors?

To confirm the immediate effects of monaural plugging before training, we studied absolute and signed errors using separate linear mixed-effects (LME) models with LISTENING (binaural or monaural) and AZIMUTH as fixed effects, and PARTICIPANT (intercept and slope) as random effect. The main effect of LISTENING (absolute error: $\underline{X^2}$ (1) = 882.70, $\underline{p}$ < 0.001; signed error: $\underline{X^2}$ (1) = 785.57, $\underline{p}$ < 0.001) and the interaction between LISTENING and AZIMUTH (absolute error: $\underline{X^2}$ (1) = 91.26, $\underline{p}$ < 0.001; signed error: $\underline{X^2}$ (1) = 57.39, $\underline{p}$ < 0.001) reached significance. Compared to binaural listening, errors increased after ear-plugging (mean ± SD; absolute error: binaural = 4.3˚ ± 4.8˚, monaural = 20.9˚ ± 11.3˚, Cohen's $d_{av}$ = 2.06; signed error: binaural = -0.2˚ ± 2.2˚, monaural = -17.4˚ ± 14.2, Cohen's $d_{av}$ = 2.14). This was particularly evident for targets delivered more toward the plugged side (see plots of absolute and signed error in Fig 2A and 2B).

Monaural plugging affected sound localization also in elevation (not shown in Fig 2; but see S1 Fig). We entered absolute and signed errors in elevation in separate LME analyses with LISTENING and ELEVATION as fixed effects, and PARTICIPANT (intercept) as random effects. We

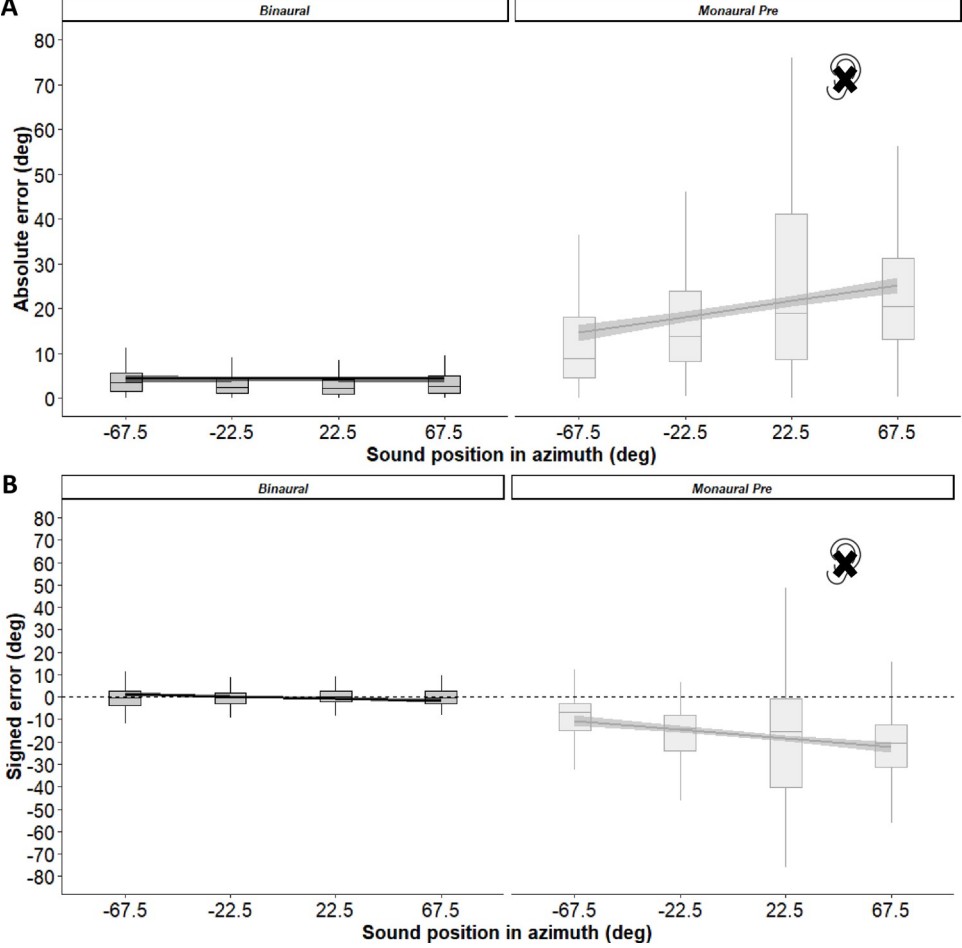

**Fig 2. Head-pointing sound localization in session 1 before training.** (A) Absolute error and (B) signed error, as a function of sound position in azimuth and listening condition. Dashed lines indicate the 95% confidence intervals of the overall linear regression. Sound positions have been grouped as a function of azimuth, irrespective of elevation.

found a main effect of LISTENING (absolute error: $\underline{X}^2$ (1) = 13.33, $\underline{p}$ < 0.001; signed error: $\underline{X}^2$ (1) = 104.6, $\underline{p}$ < 0.001). Localization errors in elevation changed after ear-plugging compared to binaural listening (mean ± SD; absolute error: binaural = 10.6 ± 6.0, monaural = 9.4 ± 2.7, Cohen's $d_{av}$ = 0.28; signed error: binaural = 9.8 ± 6.6, monaural = 5.3 ± 4.3, Cohen's $d_{av}$ = 0.83).

## Does spatial training reduce sound localization errors?

Next, we turned to examine if the spatial training was effective to reduce sound localization errors caused by monaural listening. Specifically, we studied if participants adapted to monaural listening across successive trials. To this aim, we entered absolute and signed errors into separate LME models with TRIAL NUMBER as fixed effect (25). We also included PARTICIPANT (intercept and slope) and SESSION (intercept) as random effects in the model, with the latter factor added to account for the variability related to the session in which the Spatial training was completed (first or second). The analysis on absolute error revealed a main effect of TRIAL NUMBER, $\underline{X}^2$ (1) = 4.11, $\underline{p}$ = 0.04. As shown in Fig 3A (upper panel), the absolute error reduced across trials. The analysis on signed error revealed no main effect of TRIAL NUMBER emerged ($\underline{X}^2$ (1) = 0.77, $\underline{p}$ = 0.38), but it is noteworthy that that leftward bias in sound localization decreased numerically, approaching zero (see Fig 3B).

The Non-spatial task was completed with high accuracy by all participants (percent accurate: mean±SD = 99.0±1.1) already from trial 1 and throughout the training session (Fig 3C, lower panel). During the Non-Spatial training participants were also faster in completing the trial compared to the Spatial training (Non-Spatial: mean±SD = 1.61±0.34 seconds; Spatial training: mean±SD = 4.34±1.40; $\underline{t}$ (19) = 7.96, $\underline{p}$ < .001 on paired t-test; Cohen's $d_{av}$ = 3.14).

## Does Spatial training effects generalize to the head-pointing sound localization task?

Having documented that the spatial training improved sound localization, we tested our key hypothesis about generalization of training effects to other sound localization tasks. Fig 4A shows the progression of absolute localization errors in head-pointing sound localization across the two sessions of the experiment, separately for the testing sessions before training (Pre) and after training (Post). Participants who underwent the Spatial training on session 1 and those who underwent the same Spatial training on session 2 (i.e., started with the Non-spatial training instead) are indicated by separate lines (dashed vs. continuous line, respectively). Three aspects are clearly visible in Fig 4A: (1) both trainings improved performance; (2) the Spatial training improved performance to a greater extent compared to the Non-Spatial training; (3) the interval between the two testing sessions (session 1 and session 2) made the two groups again comparable in the pre-training session of session 2, suggesting a partial wash-out of training effects.

To directly compare the effects of training type on head-pointing sound localization, we entered absolute and signed errors in separate LME models with PHASE (Pre or Post Training), TRAINING (Spatial or Non-spatial) and AZIMUTH as fixed effects. As before, we included PARTICIPANT (intercept and slope) and testing SESSION (intercept) as random effects in the model.

The results of these analyses are shown in Fig 4B and 4C. We found a main effect of PHASE, caused by performance improvements after both training types (absolute error: $\underline{X}^2$ (1) = 76.45, $\underline{p}$ < 0.001; signed error: $\underline{X}^2$ (1) = 58.07, $\underline{p}$ < 0.001), and for absolute error only an interaction between PHASE and AZIMUTH ($\underline{X}^2$ (1) = 7.78, $\underline{p}$ = 0.005). Critically, the 2-way interaction between PHASE and TRAINING also reached significance (absolute error: $\underline{X}^2$ (1) = 19.18, $\underline{p}$ < 0.001; signed error: $\underline{X}^2$ (1) = 27.26, $\underline{p}$ < 0.001; for the complete summary of results of the LME analyses see

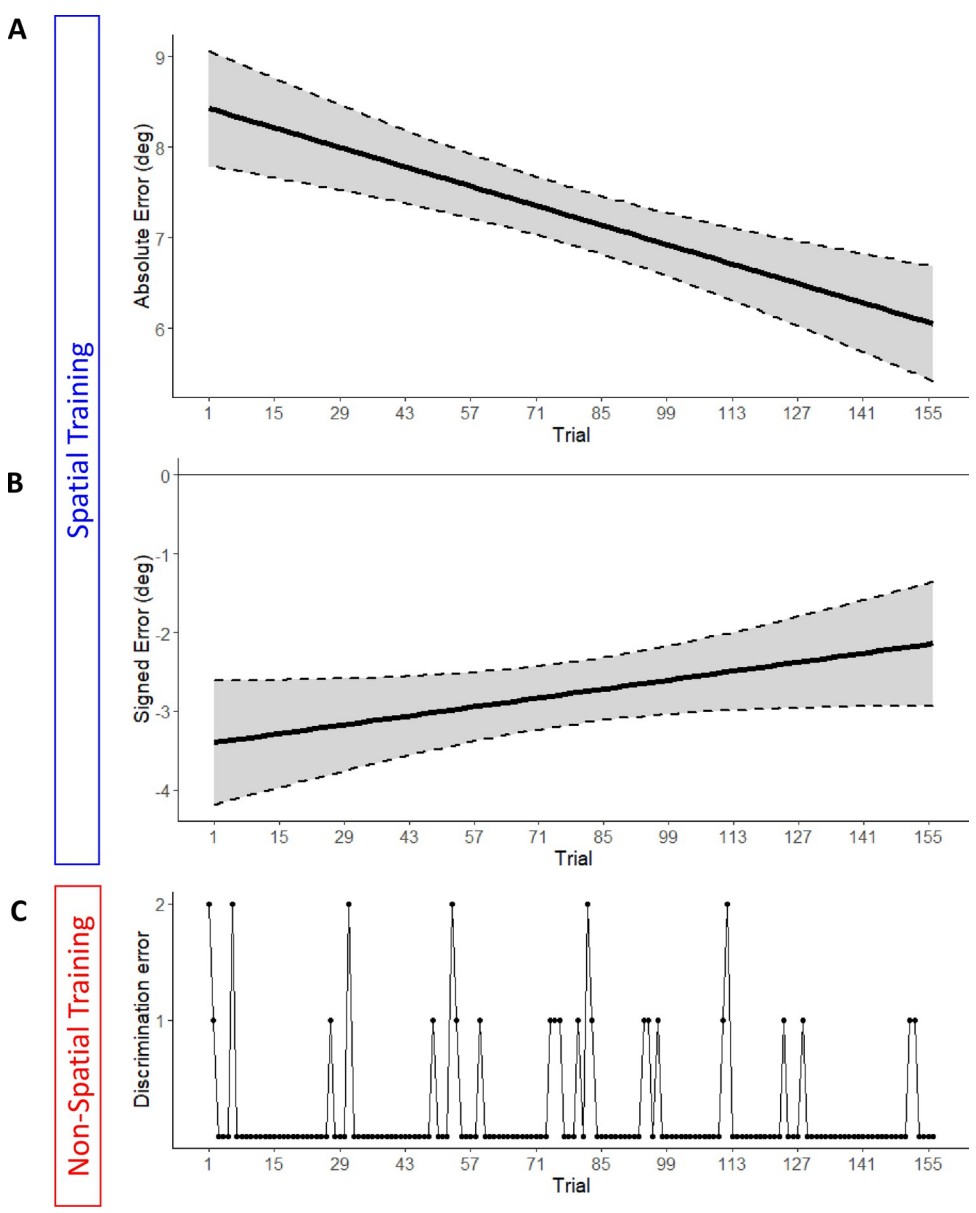

**Fig 3. Performance during Spatial and Non-Spatial training.** Reduction of absolute (A) and signed (B) error as a function of trial in the Spatial training. (C) Cumulative discrimination error across participants (i.e., number of participants who made a mistake in the trial), shown as a function of trials in the Non-Spatial training. Negative values of signed error indicate a bias toward the unplugged side. Dashed lines indicate the 95% confidence intervals of the overall linear regression.

S1 Table). Before training, localization errors were comparable across training type (absolute error: $\underline{t}$ = 0.38, $\underline{p}$ = 0.70; signed error: $\underline{t}$ = 0.23, $\underline{p}$ = 0.82). After training, errors decreased more substantially after the Spatial compared to the Non-Spatial training (absolute error: $\underline{t}$ = 5.81, $\underline{p}$ < 0.001; signed error: $\underline{t}$ = 7.61, $\underline{p}$ < 0.001). For signed error, also the 3-way interaction reached significance ($\underline{X}^2$ (1) = 7.18, $\underline{p}$ = 0.007): the impact of azimuth sound position on localization responses (i.e., larger leftward biases for sounds more toward the plugged side) decreased after the Spatial training ($\underline{X}^2$ (1) = 23.26, $\underline{p}$ < 0.001) but not after the Non-spatial training ($\underline{X}^2$ (1) = 0.24, $\underline{p}$ = 0.62). Means with standard deviations for each condition are shown in Table 1,

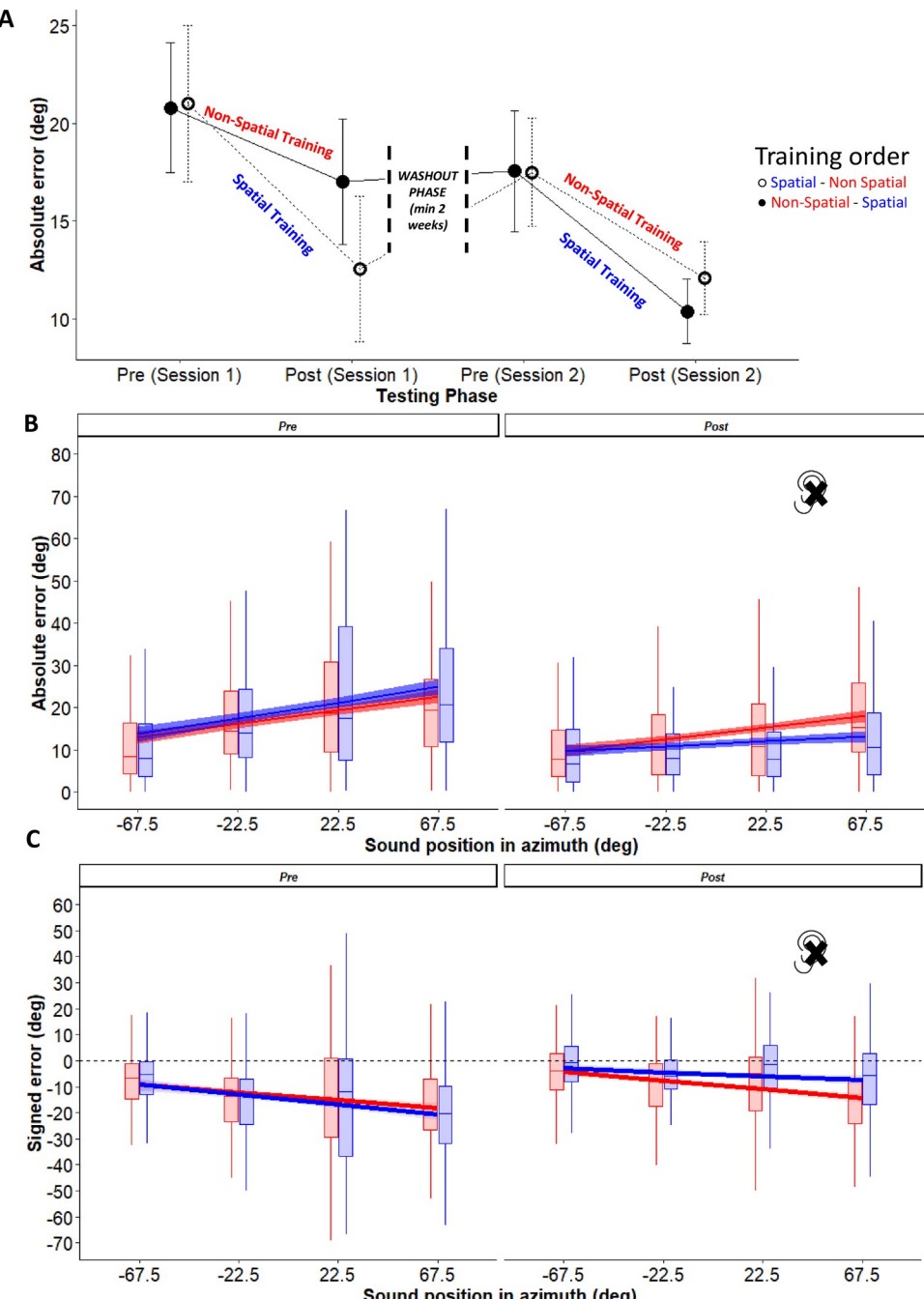

**Fig 4. Performance in head-pointing sound localization.** Top: (A) Progression of absolute localization across the four testing sessions of the experimental design, separately for participants who completed the Spatial training on session 1 (open circles and dashed line) or session 2 (filled circles and continuous line). Bottom: Absolute (B) and signed (C) errors (in degrees) in monaural listening are plotted as a function of sound position in azimuth, separately for each training type (blue: Spatial training; red: Non-spatial training). For the signed error negative values indicate a bias toward the unplugged side. Dashed lines indicate the 95% confidence intervals of the overall linear regression. Sound positions have been grouped as a function of azimuth and irrespective of elevation.

**Table 1. Mean ± SD and Cohen's $d_{av}$ for absolute and signed errors in azimuth and elevation as a function of PHASE (pre or post training) and training (Spatial or Non-Spatial during monaural listening).** Results are pooled irrespective of the order in which the two training tasks were executed.

| | | Spatial | | | Non-Spatial | | |
|---|---|---|---|---|---|---|---|
| | | Pre | Post | Cohen's $d_{av}$ | Pre | Post | Cohen's $d_{av}$ |
| Azimuth | Absolute error | 19.3°±11.1° | 11.5°± 8.9° | 0.78 | 19.1°±9.6° | 14.5°±8.4° | 0.51 |
| | Signed error | -15.1°±14.8° | -5.2°±11.3° | 0.75 | -15.3°±12.2° | -10.2°±10.2° | 0.46 |
| Elevation | Absolute error | 9.2°±2.3° | 9.5°±3.3° | 0.11 | 8.6°±2.9° | 9.2°±3.3° | 0.19 |
| | Signed error | 5.5°±3.5° | 6.5°±5.4° | 0.22 | 3.9°±4.1° | 6.0°±4.2° | 0.51 |

together with the corresponding Cohen's $d_{av}$ values (see S2 Table for each of the 8 sound positions).

To measure the wash-out effect, we entered absolute error during the post testing phase of session 1 and the pre testing phase of session 2 into a LME model. While errors of participants who performed the Non-Spatial training in the first session did not change after the wash-out ($t = 0.72$, $p = 0.47$), the errors of participants who performed the Spatial training in the first session increased after the wash-out ($t = 5.82$, $p < 0.001$, $X^2 (1) = 12.96$, $p < 0.001$).

Localization errors in elevation remained unchanged, irrespective of training type. We noted only an upward bias in the post training session compared to the pre-training one (see mean ± SD in Table 1 and full description of the analyses in S1 and S2 Tables).

## Do improvements during the spatial training predict generalization effects?

We asked if improvements during the spatial training (Fig 3) predicted the observed generalization effects, as measured in head-pointing sound localization (Fig 4). To this aim, we correlated performance in the two tasks. As indicator of improvement in the Spatial training, we used individual slope coefficients obtained from the LME model on absolute error (see Fig 3A). The higher the slope coefficient, the more the participant improved in performance during the Spatial training (for clarity, we expressed improvements as positive numbers by multiplying each slope by -1). As indicator of improvements in head-pointing sound localization, we calculated the error difference (z-normalised) before and after spatial training, separately for absolute and signed error (again, to express improvements in signed error as positive numbers we multiplied the decreasing bias by -1). The higher the error difference, the larger the training generalization effect.

A correlation between the two measures emerged. The more a participant improved during Spatial training, the greater the reduction in absolute ($R = -0.77$, $p < 0.001$) and signed error ($R = 0.57$, $p = 0.009$) during head-pointing sound localization (see Fig 5).

## Does Spatial training change spontaneous head-rotation behavior while listening?

As anticipated in the Introduction, in our previous work [25] we found that reaching to sounds increased head exploration movements during listening, and these head-movements correlated with improvements in sound localization. In this section, we examined if our training procedures also changed spontaneous head-movements during listening (from sound onset until the first response, but excluding the audio-visual feedback phase). To this aim, we examined three variables: (1) number of head-movements; (2) head-rotation extent; (3) head-rotation bias (positive values indicate rightward head-rotations). Recall that head-rotation extent and bias refer only to movements around the vertical axis (see Methods).

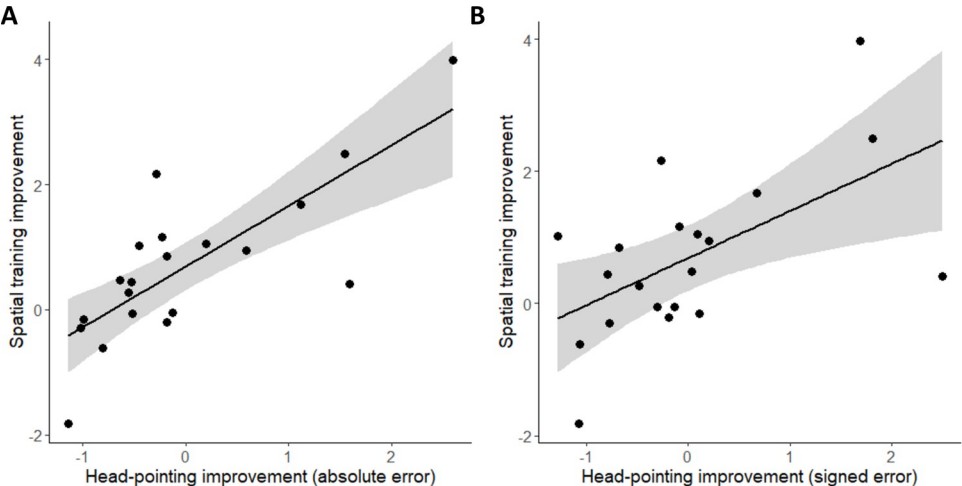

**Fig 5.** Correlation between the improvement in spatial training and the improvement in head-pointing sound localization, computed on absolute (A) or signed (B) errors.

Overall, participants made more head-movements in the Spatial (2.2 ± 0.8) compared to the Non-Spatial training (1.1 ± 0.4, $\underline{t}$ (19) = 5.94, $\underline{p}$ < .001 on paired t-test, Cohen's $d_{av}$ = 1.83). Furthermore, head-rotation extent was larger during the Spatial (64.3˚ ± 25.9˚) compared to the Non-Spatial training (4.8˚ ± 1.8˚, $\underline{t}$ (19) = 10.17, $\underline{p}$ < 0.001 on paired t-test, Cohen's $d_{av}$ = 4.29). Finally, the head-rotation bias revealed that during the Spatial training participants rotated their head more toward the right (21.3˚ ± 25.2˚) as compared to the Non-Spatial training (2.1˚ ± 4.5˚, $\underline{t}$ (19) = 3.21, $\underline{p}$ = 0.005 on paired t-test, Cohen's $d_{av}$ = 1.29). These findings indicate that spontaneous head-movements during listening were elicited more in the Spatial compared to the Non-Spatial task.

To study if and how spontaneous head-movements behaviour evolved during the Spatial training (i.e., reaching to sounds) (25), we entered the variables describing head-movements into separate LME models using TRIAL NUMBER and SIDE as fixed effect, and PARTICIPANT (intercept and slope) and SESSION (intercept) as random effects. The main effects of TRIAL NUMBER and SIDE emerged for head-rotation bias (TRIAL NUMBER: $\underline{X}^2$ (1) = 8.01, $\underline{p}$ = 0.005; SIDE: $\underline{X}^2$ (1) = 7671.55, $\underline{p}$ < 0.0001) and head-rotation extent (TRIAL NUMBER: $\underline{X}^2$ (1) = 3.82, $\underline{p}$ = 0.05; SIDE: $\underline{X}^2$ (1) = 292.83, $\underline{p}$ < 0.0001), but not for number of head movements (TRIAL NUMBER: $\underline{X}^2$ (1) = 0.003, $\underline{p}$ = 0.96; SIDE: $\underline{X}^2$ (1) = 1.27, $\underline{p}$ = 0.26). As shown in Fig 6A, participants increased their head rotation bias toward the plugged (right) side (positive) as a function of trial repetition. This head-orienting behaviour is compatible with participants exposing their unplugged ears toward the sound. For head-rotation extent, also the 2-way interaction reached significance (TRIAL NUMBER X SIDE: $\underline{X}^2$ (1) = 3.81, $\underline{p}$ < 0.05, Fig 6B). Finally, changes in head-rotation extent and head-rotation bias as a function of trials were correlated with one another ($\underline{R}$ = 0.47, $\underline{p}$ = 0.04). In sum, during Spatial training phase, participants oriented more their heads toward the plugged (right) side as training progressed. Moreover, they increased the space explored with the head, particularly when sounds were delivered on the plugged side (Fig 6A and 6B).

One final aspect worth noting concerns the relative timing of spontaneous head-movements and required hand-responses (i.e., reaching to sound) during the Spatial training. While the first head movement started on average 0.97 seconds (SD = 0.27) after sound emission, the hand-held controller touched the speaker on average 3.78 seconds (SD = 1.33) after sound emission. This indicates that head-movements implemented during listening preceded hand-

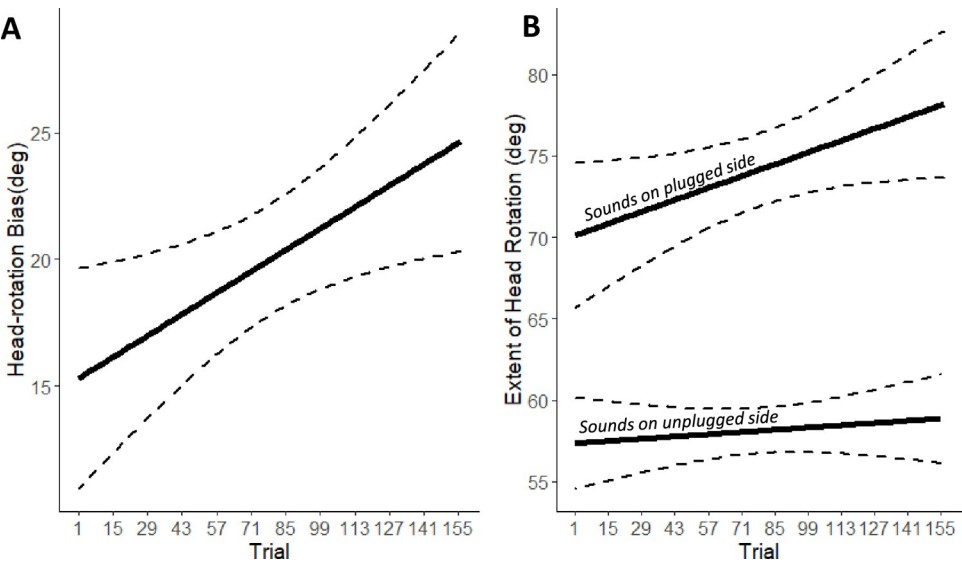

**Fig 6. Changes in head behavior during the spatial training.** Head-rotation bias as a function of trial (A) and extent of head-rotation as a function of trial and separately for sound delivered to the left or to the right of participant's midline (B). Dashed lines indicate the 95% confidence intervals of the overall linear regression.

reaching movements required to provide the response, hence were not the mere consequence of hand-head coordination towards the target sound.

## Does training effects change immediate head-orienting to sounds?

In the head-pointing sound localization task, head-movements were functional to the response, hence they cannot be considered fully spontaneous (unlike in the training tasks). Yet, the first head-movement (which occurred around 0.99 seconds after the beginning of the sound) could be taken to reflect the immediate and spontaneous orienting response toward the sound. Here, we focused on these first movements to study if training effects can be detected also in immediate head-orienting to sounds. Specifically, we examined the horizontal direction of the first head-movement and its onset (i.e., Reaction Time, RT, in seconds).

An LME model on the horizontal direction of the first head-movement (with PHASE, TRAINING and SIDE as fixed effects and PARTICIPANT as random effect) revealed that first head movements were overall directed toward the correct hemispace (right: 46.2±26.9; left: -42.0±26.6; main effect of SIDE, $\underline{X}^2$ (1) = 1270, $\underline{p}$ < 0.001). Moreover, there were more rightward oriented responses after training (6.5±25.4) compared to before training (-2.3±26.0; PHASE, $\underline{X}^2$ (1) = 8.82, $\underline{p}$ = 0.003). This indicates that head-orienting to sounds was more biased towards the right (plugged) side after training, but irrespective of training type.

Most interestingly, a similar LME model on first head-movements RT (analyzed only for the trials in which the head-movement direction was correct, 97.1%) showed a 3-way interaction between PHASE, TRAINING and SIDE ($\underline{X}^2$ (1) = 9.33, $\underline{p}$ = 0.003). This interaction is illustrated in Fig 7. Before training, sounds delivered from the plugged (right) side resulted in slower first head-movements compared to sounds delivered from the left side, for both the Spatial (1.01±0.29 vs. 0.99±0.34; $\underline{p}$ < 0.001) and the Non-Spatial (1.07±0.33 vs. 1.02±0.32; $\underline{p}$ = 0.02) training. This difference in RT latency for the plugged side reversed selectively after the Spatial training (left: 0.96±0.28; right: 0.92±0.28; $\underline{p}$ = 0.05), whereas it persisted after the Non-Spatial training (left: 1.02±0.32; right: 1.03±0.28; $\underline{p}$ = 0.009). This finding provides further

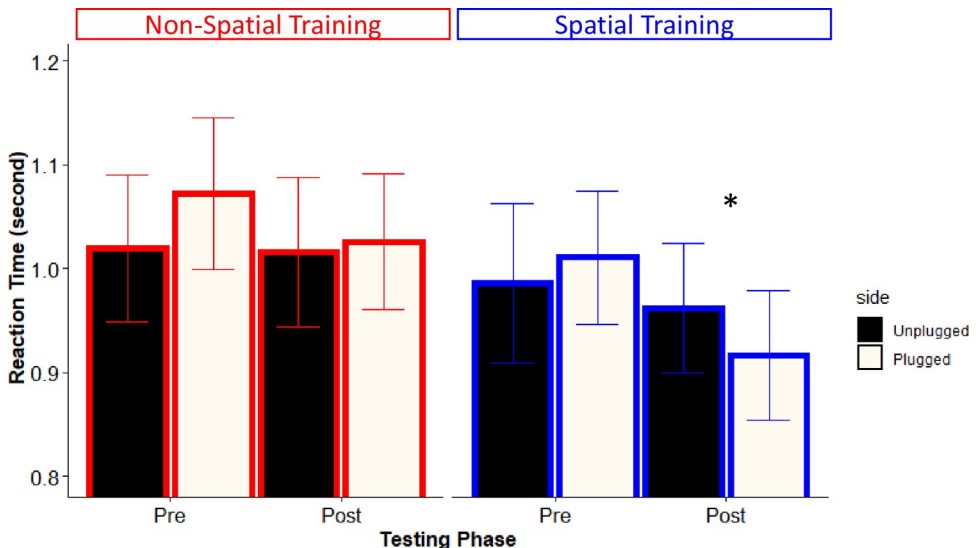

**Fig 7. Reaction Time, RT, in seconds as a function of training type (Non-Spatial or Spatial) phase (pre and post training), separately for sound delivered to the left (black) or to the right (white) of participant's midline.** Error bars indicate the standard errors. In all comparisons RTs for right targets are significantly slower than those for left targets, except in the post phase of the Spatial training. In the latter case the RT pattern is significantly reversed (marked by an asterisk).

support to the notion that participants developed a bias in orienting their heads toward the right (plugged) hemispace, specifically after the Spatial Training.

## Does training induce changes in audio-visual attention orienting?

Finally, we turned to investigate the effects of training in the audio-visual attention cueing task to examine if generalization effects emerged also for this implicit spatial hearing task. To assess the immediate effects of monaural plugging on the audio-visual cuing task, we compared attention Cueing Effects (CE, in milliseconds) in binaural and monaural listening during the first session. The larger the CE the more participants exploited the lateralised sound as a cue for visual attention, i.e., they implicitly took advantage of its spatial position. We entered CE values in a LME model with the LISTENING (binaural or monaural) as fixed effect and PARTICI-PANT (intercept) as random effects. As expected from previous works [26], we found a main effect of LISTENING ($\underline{X}^2$ (1) = 12.24, $\underline{p}$ < 0.001) revealing that the CE decreased after monaural plugging (binaural = 20.2 ± 11.0 ms, monaural = 6.8 ± 13.1 ms, Cohen's $d_{av}$ = 1.11).

To study the effects of our training protocols, we entered the CE values in monaural listening conditions into a LME model with PHASE (Pre or Post training), TRAINING (Spatial or Non-Spatial) as fixed effects, and PARTICIPANT and SESSION (intercepts) as random effects. No main effect or interaction emerged from this LME model (all $\underline{ps}$ > 0.49). As shown in Fig 8, CE remained unchanged between pre and post testing phases, in both the Spatial (pre = 6.9 ± 9.5 ms, post = 9.1 ± 14.6 ms) and Non-Spatial training (pre = 6.0 ± 16.1 ms, post = 3.9 ± 15.9 ms). This indicates that our spatial training affected performance during the training task itself, during head-pointing to sounds (both in terms of localization errors and first head-movements), but did not improve the implicit processing of sound side in relation to the visual stimulus.

## Discussion

Generalization is essential when assessing the potentials of any training procedure [10]. Here we tested if a training based on sound-oriented actions can result in adaptations that extend

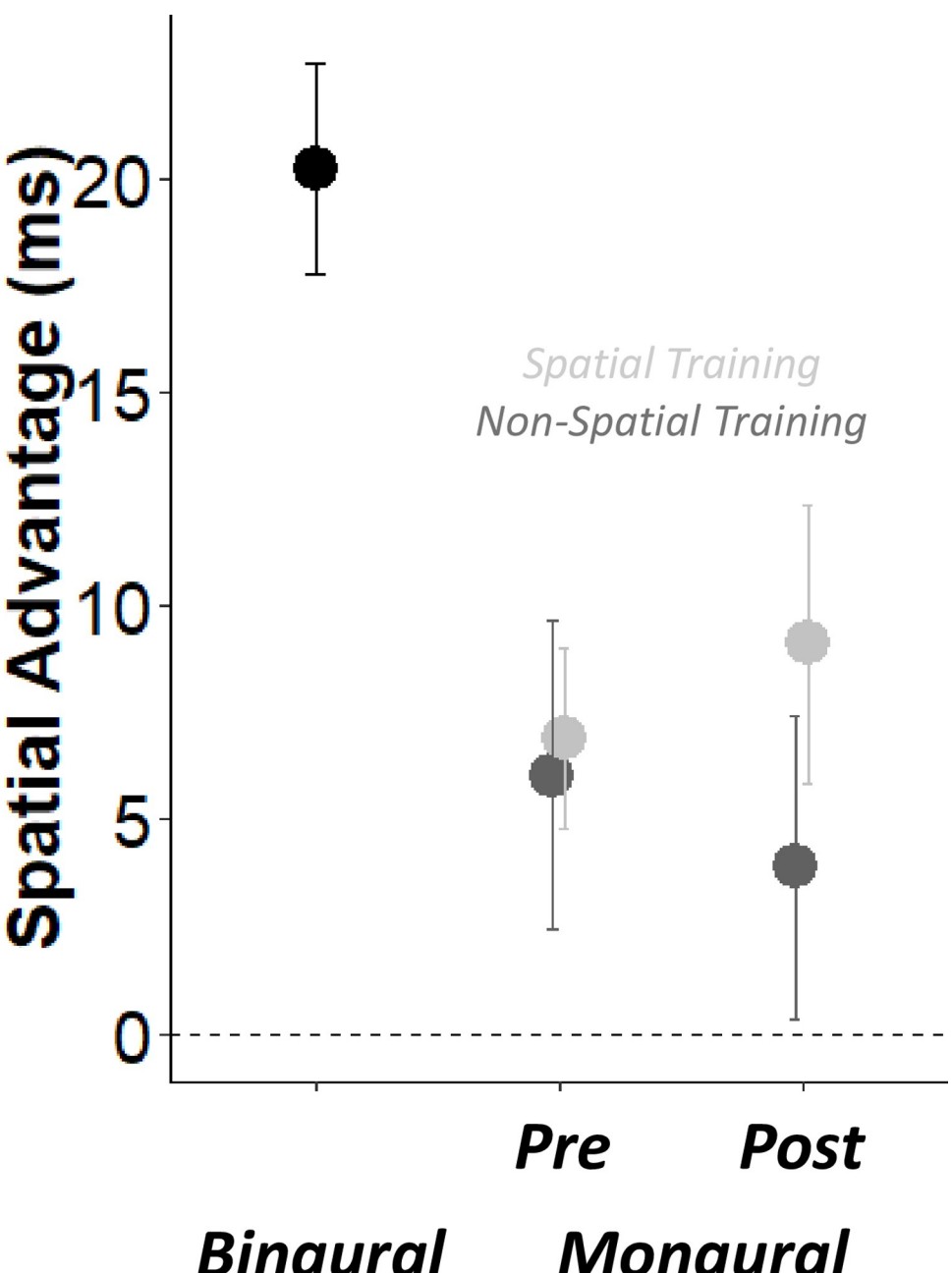

**Fig 8. Audio-Visual Cueing task.** Audio visual cueing (ms) as a function of PHASE and TRAINING (Spatial in light grey and Non-Spatial in dark grey).

(i.e., generalize) to other untrained sound localization tasks. Using a crossover experimental design, we examined the effects of a Spatial training (indicate sound position through reaching) on two tasks performed before and after the training session: a head-pointing to sound task and an audio-visual attention cueing task. As a control, we used a Non-Spatial training (indicate sound type through reaching). Three main findings emerged. First, we confirmed that listeners using an ear plug can rapidly improve sound localization while performing a Spatial training based on reaching to sounds [25]. Second, we found that the improvements

induced by the Spatial training, generalize to head-pointing sound localization. Instead, the audio-visual attention cueing task remained unaffected. Third, we documented changes in head-movement behaviour during the Spatial training, and we provide initial evidence that such head-movement adjustments can transfer to an untrained auditory spatial task. In the next paragraphs, we discuss each of the main findings in turn.

## Reaching to sound is an effective training strategy

In this study, we observed generalization of training effects, particularly after the Spatial training. These results confirm and extend previous findings documenting the benefits of training procedures based on audio-visual feedback [11, 12] and motor interactions with sounds [13]. In these previous works, however, the efficacy of training protocols was always examined using between-subject experimental designs that grouped participants as a function of the proposed training type. By contrast, in the present work, we successfully used a crossover design to compare generalization effects of two training procedures within-subjects. This design gave us the opportunity to minimize any potential intergroup differences. Furthermore, such a crossover design makes this paradigm particularly suited for clinical applications, because it permits to involve the totality of participants in the experimental training without the need of testing a control group (which could not benefit from training).

Our Spatial training approach took advantage of the benefits of audio-visual feedback related to sound position [as in 11 and 12]. In addition, it was characterised by the presence of goal-directed actions toward each sound. We built on works in which interactions with sound sources occurred using reaching movements, as in some pioneers VR approaches [21, 22] and more recent studies [23]. A motor approach based on hand movements in the space of the sound sources has also been adopted by Valzolgher and colleagues [13], in a study that demonstrated the effect of kinesthetic cues when re-learning to localize sounds with one ear plugged, as here. In the above-mentioned previous work, however, the action was not sound-oriented and it was largely repetitive: participants were instructed to move their arm repeatedly over the speakers while wearing a sound-emitting bracelet attached to their wrist. Conversely, in the Spatial training proposed in the current study, participants performed a reaching to sounds action. Our working hypothesis was that reaching to sound could enhance spatial coding of sound position by favouring the coordination of different effectors (eyes, head, hand) into a common reference frame [25, 43, 44]. In addition, reaching to sounds could help directing attention toward the position occupied by the sound source and make the task more engaging, which is a fundamental feature in any learning procedure [45].

While the task requirement in our Spatial training was to reach each sound source with the hand, one critical feature was that participants were free to make spontaneous head-movements while listening to the sounds. The rationale for introducing this spontaneous behaviour in our training was based on two notions. First, head movements imply continuous updates in the acoustic cues reaching the ears, which may reduce ambiguities in the altered auditory cues and, eventually, favour re-learning of sound-space correspondences [46–48]. Second, by moving the head listeners can discover and implement self-regulation strategies which may prove adaptive when coping with the altered auditory cues. In the present study, for instance, a clear strategy emerged: the participant moved the head to expose the unplugged ear to the sounds. Steadman and colleagues [24] also tested three training approaches based on head pointing to sound and audio-visual feedback. Interestingly, their training based on active listening, in which head movements were permitted during sound playback, resulted in greater improvements in localization accuracy, compared to the two other training they proposed (i.e., a non-gamified and gamified version of the training in which the head remained still). In their study,

the time for exploring the auditory scene with head movements was limited to 1.6 seconds, whereas in our experimental training listening occurred without time restriction (with an average sound duration of 4.6 seconds). This difference in sound duration may have led to greater exposure to changes in the auditory cues related to spontaneous head-movements, thus fostering more strategic behaviors to identify sound sources. In future studies, it would be interesting to understand which are the most relevant specific aspect(s) of sound involved motor interactions in our Spatial paradigm.

Although generalisation effects were greater after the Spatial than after the Non-spatial training, performance in the head-pointing task improved significantly after both training procedures. This indicates that participants spontaneously adapted to the altered listening situation, even without feedbacks about the spatial position of sounds (recall that the audio-visual feedback in the Non-Spatial training only entailed the categorical discrimination between the two sound types). Improvements in the absence of relevant training or even no training at all (i.e., simple test-retest) have already been documented in literature [11, 49]. In the present paradigm several methodological aspects could have favoured this spontaneous relearning. First, sounds were emitted from 8 different positions during the head-pointing task and participants may have learned them between the pre- and the post-training sessions, although no visual cues about sound location were ever provided. Second, participants may have become more familiar with the task and with the altered auditory experienced. The performance improvements observed irrespective of training type, however, may be qualitatively different compared to the ones observed after the Spatial training. Indications in this direction may emerge from the differential effects of the wash-out phase as a function of the training performed in the first session (see Fig 4A). The wash-out phase was effective for participants who performed the Spatial training during the first session, whereas it did not change performance for participants who performed the Non-Spatial training. This observation may distinguish between two types of improvement: a more stable one, common to both groups and possibly related to the generic experience with the auditory alteration and the experimental setup, and more a contingent one, proper to the features of the Spatial training.

An aspect worth discussing concerns the lack of training effects in the vertical dimension. In our study the ear plug effect affected sound localization specifically in the horizontal dimension, and altered monaural cues (required to localize in elevation) only at the plugged ear. In addition, the Spatial training protocol provided a rich visual feedback only for the horizontal position of sounds, whereas the only information about sound elevation was related to the continuously visible array of speaker that showed the veridical sound height in both training types. Finally, the difference in elevation (10°) in the head-pointing task was introduced primarily as a way to increase uncertainty about sound position and to differentiate between sound positions in the trained and untrained auditory spatial tasks. Note that the reduced range of elevation (20 degrees) made any potential improvement difficult to see, compared to the larger range of azimuths. It would be interesting to examine if our Spatial training could also prove effective to improve sound position in elevation when hearing is altered using earmoulds [as in 50] or by degraded spectral content [as in 49].

One word of caution concerning our methodology relates to the different length of the two training protocols. While participants were fast in identifying the amplitude modulation rate of target sounds (mean trial duration in the Non-Spatial training was 1.6 seconds), identification of their spatial source required a longer listening time (mean trial duration in the Spatial training was 4.3 seconds). This implies that two factors could have contributed to the Spatial training efficacy we have documented: the spatial task requirements and the longer stimulus duration. Future studies could overcome this methodological limitation by creating tasks of comparable difficulty, or by imposing a fix sound duration in each trial. Note however that

extending trial duration in the Non-Spatial task could allow participants to process both the sound's amplitude modulation rate and the sound's spatial position, thus making task-relevant and task-irrelevant sound feature more difficult to disentangle.

### A possible role for head-movements in adapting to altered spatial hearing

Although studies regarding the contribution of head-movements to spatial hearing have been advocated since the 1940s, only few works have examined how head-movements could change while adapting to altered auditory cues. Our VR approach to spatial hearing made it possible to study head movements throughout the experiment and we always presented sounds with long durations (i.e., until response, during the training phase; or 3 seconds long, during the head-pointing task) precisely to examine the contribution of head-movements during training and generalization. Our work adds to the limited body of work that focused on spontaneous head-movements during sound localization relearning [46–48, 51] by contributing two main findings.

The first finding emerged during the Spatial training phase. While adapting to the altered auditory cues, participants increased head-rotation extent to explore a wider portion of space during listening. Furthermore, they turned their heads more to the right as training trials progressed, as revealed by the increasing head rotation bias. These two head-orienting behaviours can be conceived as spontaneous adaptation strategies for coping with the altered binaural cues, which change spatial hearing specifically in the horizontal dimension. In particular, the rightward bias in head-orienting allowed listeners to direct their unplugged (left) ear toward the sound sources, even when sounds originated from the plugged side. The second finding emerged in the post-training phase. Even in the untrained task (i.e., head-pointing to sounds), we observed again a bias in exploring the right (plugged) side of space. After training, participants directed their first head-movement (within the first second from sound onset) more to the right. Moreover, they triggered this rightward movement faster specifically after the Spatial training. This suggests that the head rightward bias implemented during the Spatial training may have transferred, at least to some extent, to the untrained task.

One important aspect to notice is that head movements during the Spatial training were not only functional to orient the head to the target selected for reaching. Actually, most head-movements preceded reaching, hence they were spontaneously implemented by participants. Although 61.5% of trials fell within 10 degrees from the speaker reached with the hand, in the remaining proportion of trials the head was not aligned with the hand. In fact, the misalignment between the hand and the head was 9.7 degrees (SD = 8.7) when the hand reached the target speaker. Moreover, during the Spatial training, participants made 2.2 movements (on average) in each trial, and the timing of head-movements and reaching actions differed. The first head movement during the Spatial training started around 1 second after the beginning of the sound, whereas the first hand-contact with the speaker in each trial occurred around 4 seconds after the beginning of the sound. Most notably, the rightward bias of the head we described above indicates that head movements were not only performed to coordinate head-pointing with the hand-reaching response. Taken together, these findings indicate that participants explored the auditory scene with their head before taking the decision to act on the sound source. One important implication of this this finding is that head-movements during the Spatial training likely served two purposes: exploration of the acoustic scene and support for the reaching response phase. The latter coupling between head-movements and reaching response reveals a partial overlap between the motor response required for the untrained task (head-pointing to sound task) and the Spatial training task (head movements that accompanied the hand reaching response). Future work should address generalization also using

auditory tasks that do not entail any such overlap in the motor response between trained and untrained task (e.g., assessing generalization through minimum audible-angle tasks, as in [13]).

To what extent spontaneous head-movements during sound contributed to the exploitation of auditory cues remain to be ascertained. As suggested above, one possibility is that head movements dynamically changed the auditory cues to location, thereby reducing ambiguity in the incoming signals. This better sampling of the auditory cues under the altered listening condition could have then favoured the creation of new correspondences between cues and space. A second possibility, however, is that head-orienting did not serve the purpose to reduce ambiguity, but allowed to better exploit monaural signals at the unplugged ear. Specifically, participants may have learned to use sound intensity at the unplugged ear as a proxy of sound position, searching head-position at which sound intensity reached a peak and then encoding the position in space in memory. Our findings are suggestive of this second head-orienting strategy, because participants clearly developed a bias in head-orienting toward the plugged side (which implies exposing more the unplugged ear to the frontal sounds). Yet, we cannot exclude that by implementing this strategy they also acquired richer information about the auditory cues and reduced their ambiguity.

A better understanding of the relations between head-movements and the exploitation of auditory cues would be important to characterize in which way the spontaneous head-orienting behaviors interact with the two cognitive mechanisms that have been typically associated with sound localization relearning, namely 'cue-reweighting' and 'cue-remapping'. Cue-reweighting, indicates the mechanism by which the brain changes the relative weight attributed to each auditory cue (ILD, ITD and monaural), giving greater prominence to the reliable auditory cues at the expenses of the more impoverished ones [52]. For instance, when one ear is plugged the brain can learn to weight monaural cues more than the altered binaural cues, when localizing sounds in azimuth. Instead, cue-remapping [53] refers to the ability to create new correspondences between auditory cues and spatial positions through experience. Although all these mechanisms could have played a role in the observed sound localization improvements, we cannot examine exactly the extent of their relative contribution primarily because of our use of naturalistic stimuli in the free-field and the adoption of head-movements during listening in most tasks. Future studies should address in which way allowing spontaneous head-movements during training could impact on cue-reweighting and cue-remapping mechanisms, for instance by pairing our active Spatial training task with more controlled auditory inputs to the ears and tasks performed with the head static.

## No generalization of training to the audio-visual attention cueing task

In the present work we also attempted to examine if training acoustic space perception could have effects beyond explicit sound localization alone. To this aim, we tested generalization effects in an audio-visual attention cueing task, in which sound localisation skills are functional to capture attention to the portion of space in which a visual target appears. However, our findings did not reveal any generalization to this task.

While negative findings must be interpreted with great caution, one methodological difference between the audio-visual attention cueing task and the head-pointing task is worth noting. Head movements were entirely prevented by a chin-rest in the attention cueing task, whereas they were fully allowed in the head-pointing task. This aspect was not the only difference between the two tasks: they were also conducted outside vs. inside VR, they entailed indirect vs. direct used of auditory spatial information, respectively. Yet, when considered together with the observations reported above on the prominence of head-movements during training

and the possible role of this behaviour in the generalization effects, restrained head-movements may have been a critical factor limiting generalization of training to our attention task.

## Conclusions

Our findings show that reaching to sounds improves spatial hearing in participants using an ear-plug, more than a control condition using the same stimuli but non-spatial task demands. Training by reaching also modified head-movement behaviour. Crucially, the improvements observed during training generalized to a different sound localization task, possibly as a consequence of acquired and novel head-movement strategies. These findings extend the current perspectives about the mechanisms by which humans can adapt to altered auditory cues by showing the important role of active interactions with sound sources (reaching and head-movements) during relearning. Note that these interactions are prevented when participants use a chin-rest, and may be limited when sounds are delivered for a very short time and beyond reaching. In this respect, our results point to a methodological shift towards more active listening scenarios, which could better approximate the naturalistic experience with sounds. They also have implications for clinical and applied settings because they demonstrate that implementing multisensory-motor approaches to train acoustic space perception can be done effectively. Future studies could extend this approach to hearing-impaired population with spatial hearing difficulties. For example, they could test the effectiveness of reaching-to-sound training protocols by measuring the performance of deaf patients with unilateral hearing loss or people who wear cochlear implants or hearing aids [54].

## Supporting information

**S1 Fig. Head-pointing sound localization in elevation.** Density plot of responses during Session 1, as a function of sound position in elevation. The center of the speaker was located at -15 or +5 with respect to the ear level (shown by dashed lines), depending on the trial. This asymmetrical arrangement of the speakers resulted from the fact that the software for placing the speakers at pre-determined positions was based on the position of the VIVE tracker above the speaker, rather than the position of the actual speaker.
(PDF)

**S1 Table. Results of the LME analyses.** LME analyses of absolute and signed errors in azimuth and elevation.
(PDF)

**S2 Table. Absolute and signed errors in azimuth.** Mean ± SD for absolute and signed errors in azimuth as a function of 8 target positions and PHASE (pre or post training) and training (Spatial or Non-Spatial during monaural listening).
(PDF)

## Acknowledgments

We are grateful to all the participants in the study and to Dr. Stefania Benetti for comments on a previous version of the manuscript. We thank Isabelle Viaud-Delmon and one anonymous reviewer for their comments and suggestions on a previous version of the manuscript. We thank Jean-Louis Borach and Sonia Alouche for administrative support of this project and Eric Koun for technical support. We thank Giordana Torresani for her contributions to Fig 1 and Angela Valenti for improving English in a previous version of the manuscript.

## Author Contributions

**Conceptualization:** Chiara Valzolgher, Alessandro Farnè, Francesco Pavani.

**Data curation:** Chiara Valzolgher, Julie Gatel.

**Formal analysis:** Chiara Valzolgher.

**Funding acquisition:** Chiara Valzolgher, Eric Truy, Alessandro Farnè, Francesco Pavani.

**Investigation:** Chiara Valzolgher, Michela Todeschini, Valerie Gaveau.

**Methodology:** Chiara Valzolgher, Alessandro Farnè, Francesco Pavani.

**Project administration:** Chiara Valzolgher, Julie Gatel, Eric Truy, Alessandro Farnè, Francesco Pavani.

**Resources:** Gregoire Verdelet, Julie Gatel, Romeo Salemme, Eric Truy.

**Software:** Gregoire Verdelet, Romeo Salemme.

**Supervision:** Alessandro Farnè, Francesco Pavani.

**Visualization:** Chiara Valzolgher, Francesco Pavani.

**Writing – original draft:** Chiara Valzolgher, Francesco Pavani.

**Writing – review & editing:** Chiara Valzolgher, Valerie Gaveau, Alessandro Farnè, Francesco Pavani.

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
