## [Decision Letter · Decision Letter 0]

7 Oct 2021

PONE-D-21-25716Adapting to altered auditory cues: generalization from manual reaching to head pointingPLOS ONE

Dear Dr. Valzolgher,

Thank you for submitting your manuscript to PLOS ONE. After careful consideration, we feel that it has merit but does not fully meet PLOS ONE’s publication criteria as it currently stands. Therefore, we invite you to submit a revised version of the manuscript that addresses the points raised during the review process.

 ==============================I agree with the reviewers that your submission has merit but also agree with them that some points require revision. It is my opinion that you will be able to answer the main points raised by the reviewers promptly. Of course, your responses will be assessed by myself and the same reviewers once submitted.==============================

We look forward to receiving your revised manuscript.

Kind regards,

Welber Marinovic

Academic Editor

PLOS ONE

“The study was supported by a grant of the Agence Nationale de la Recherche (ANR-16-CE17-0016, VIRTUALHEARING3D, France) to F.P. and A.F. In addition, C. V. was supported by a grant of the Università Italo-Francese/Université Franco-Italienne, the Zegna Founder's Scholarship and Associazione Amici di Claudio Demattè. F. P. and A.F. were also supported by a prize of the Foundation Medisite (France), by the Neurodis Foundation (France) and by a grant from the Italian Ministry for Research and University (MUR, PRIN 20177894ZH). Finally, A.F. was supported by the IHU CaSaMe ANR-10-UBHU-0003 and ANR 2019CE37 Blind Touch. The funders had no role in study design, data collection and analysis, decision to publish, or preparation of the manuscript.”

“The study was supported by a grant of the Agence Nationale de la Recherche (ANR-16-CE17-0016, VIRTUALHEARING3D, France, https://anr.fr/) to F.P. and A.F. In addition, C. V. was supported by a grant of the Università Italo-Francese/Université Franco-Italienne (https://www.universite-franco-italienne.org/), the Zegna Founder's Scholarship (https://www.zegnagroup.com/it/csr/founder-scholarship/) and Associazione Amici di Claudio Demattè (http://www.amicidematte.org/). F. P. and A.F. were also supported by a prize of the Foundation Medisite (France), by the Neurodis Foundation (France) and by a grant from the Italian Ministry for Research and University (MUR, PRIN 20177894ZH) (https://www.miur.gov.it/). Finally, A.F. was supported by the IHU CaSaMe ANR-10-UBHU-0003 and ANR 2019CE37 Blind Touch (ANR: https://anr.fr/). The funders had no role in study design, data collection and analysis, decision to publish, or preparation of the manuscript.”

Additional Editor Comments (if provided):

Reviewers' comments:

Reviewer's Responses to Questions

**Comments to the Author**

1. Is the manuscript technically sound, and do the data support the conclusions?

Reviewer #1: Partly

Reviewer #2: Yes

2. Has the statistical analysis been performed appropriately and rigorously? 

Reviewer #1: Yes

Reviewer #2: Yes

3. Have the authors made all data underlying the findings in their manuscript fully available?

Reviewer #1: Yes

Reviewer #2: Yes

4. Is the manuscript presented in an intelligible fashion and written in standard English?

Reviewer #1: Yes

Reviewer #2: Yes

5. Review Comments to the Author

Reviewer #1: This study examined the effect of spatial and non-spatial auditory training on the localisation of sounds and on auditory cueing of visual stimuli. Spatial training improved spatial localisation, but not spatial cueing. Lower improvements were seen for non-spatial training. Importantly, the authors report that spatial training with arm reaching generalised to localisation with movements of the head, suggesting integration of auditory, arm and head coordinate frames. However, it is not clear to what extent participants moved their head, as well as their arms, during training. Conclusions about generalisation are therefore difficult to draw. Also, the authors point to the significant effect of spatial training on localisation, but not on cueing. The statistics for cueing are only cursorily described and the data are relegated to the Supplementary Material. To what extend are these results important to the conclusions of the study? They could be more visible and more thoroughly discussed.

Specific comments

Head-pointing localisation task: While it is true that head movements made during sound presentation will dynamically change auditory cues to location, thereby reducing ambiguity, head-pointing to sounds is a special case where interaural cues can be nulled. The ear plugged conditions are only monaural to the extent that the plug/muff combination sufficiently attenuates sound. To what extent would interaural time differences still be available at low frequencies where plugs/muffs are less attenuating? Could it be that participants are able to null residual ITDs?

Line 74. Head movements and localisation accuracy were both higher in the reaching group than in the naming group. Can you be sure that it was arm reaching and not head movements that caused the improvement in localisation?

Figure 2 is duplicated as Figure 3. Figure 3 is missing.

As I understand it, the linear mixed effect model estimates the linear regression of error on azimuth, with listening condition and participant as fixed and random factors. Is there any reason to assume that the relationship of error and azimuth is linear? The rationale for examining the linear trend across azimuth could be better articulated.

Figure 2C: Can you speculate why localisation bias changed across source azimuth in the monaural condition? For each location, the head would have been more-or-less centred on the source at time of response. If localisation cues were head-centred, one would have expected similar results across source locations.

Figure 4A legend appears to be reversed.

Line 439. To what extent are the head movements spontaneous in the training task? Are they not related to the task where the participant must visually guide the controller to the response location?

There were only 8 locations (4 azimuths and 2 elevations) in the testing phase. Did participants learn these locations over test trials? Localisation performance is different when from a limited set of alternatives than from an open set.

Line 506. Why were Phase and Training included as random effects?

Discussion:

Line 514. “that differ in terms of both stimuli and motor response.”

The stimuli for both training and test sessions was amplitude modulated white noise. The motor response for the test session was a head point to the target. The motor response for the spatial training was an arm reach, but it not clear to what extent participants also moved their heads. If they oriented toward the response location, then motor responses would be similar between training and test sessions, with the exception that training also required the arm reach. It is only if orienting head movements were not allowed during training that one can make a case for generalisation from arm reaching in training to head pointing in test.

Line 601. An argument is made that changes in head movement may be a form of cognitive unloading due to the perceptually complex nature of the task. The rationale for this possibility is not well articulated. Head movements resolve ambiguity, but it is not clear that represents cognitive unloading.

The English expression is generally good, but there are a number of places that would benefit from proof reading, for example plurals expressed as singular etc.

Reviewer #2: The aim of this study about adaptation to altered auditory cues is at least three-fold. It addresses whether a spatial auditory training procedure requires explicit or implicit processing of sound localization to be efficient; whether the training procedure resting on 13 specific sound positions (13 different azimuths) has also an impact on non-trained positions of the auditory space (4 different azimuths and 2 different elevations); and whether the spatial training based on reaching to sound is also efficient on localization performances through head orienting to sound.

This is a very nice study, carefully conducted, clearly written and presented.

I have only minor comments, that might help to make the message clearer.

Page 1, Author Note

Correct the typo in “Data can be retrieved from osf.io/dt76.” Since it is osf.io/dt76h as indicated later in the text.

Abstract:

“When the auditory cues change, sound localization becomes difficult and uncertain.”

This sentence is not easy to understand. Add “when auditory cues change, because of…or…., sound localization becomes ….”. Otherwise we do not grasp the meaning, since auditory cues change constantly.

Highlights:

It would be nice to modify the last one to add how the findings can be applied to rehabilitation protocols.

In the section “Procedure and stimuli”, please indicate the approximate duration of each of the different phases of the protocol. I could find it only for the AV cueing task.

Line 297 and 304, you mention that trials were removed from the analysis in the head pointing tasks and in the AV cueing task (incorrect responses). Please indicate for each task how many trials have been removed.

FIGURE 1: modify illustration C for more clarity. There is an arrow only on the right towards the non spatial training. Make sure that the illustration of the setup shared by the two types of training is in the middle of the figure.

FIGURE 2: B and C. the absolute and signed errors are represented as a function of sound position in azimuth, but it is not straightforward to understand since the indicated azimuths on the x axis are not those that have been tested but only theoretical points (same for figure 3 and 4). Please also explain that the two tested elevations are cumulated for the analysis, taking into account azimuth only.

Line 388 “The interval between the two testing sessions was effective in inducing a partial wash-out of training effects.”

Could you please discuss further this “partial” wash-out? How could you explain that the group with non-spatial training in the first session remained stable while the group with spatial training had a higher mean absolute error in the pre-test of session 2? Would it be relevant to do at the beginning of each session a binaural baseline?

Line 411 “Localization errors in elevation remained unchanged, irrespective of training type.”

This finding is not discussed further, while it might be interesting. Might this difference be linked to the different mechanisms involved in the processing of binaural difference cues and in the processing of spectral cues? Monaural plugging alters dramatically binaural difference cues, but leaves unchanged monaural spectral cues on the remaining ear. Several findings suggest that those cues have an independent role in auditory localization (see for example Sarlat et al 2006). Alternatively, you might of course consider that the difference in elevation (10°) is so small that the absolute error exceeds the size of the difference between the two positions.

Table 1: it would be nice to see somewhere the data for all the 8 stimuli instead of means of the 8 positions. Maybe it could be added in supplementary data?

Line 507 Cueing effect: no main effects or interactions emerged. But on figure 7, and in its legend, an effect is marked. Please explain this effect in the text and comment on it. Indeed, on line 521 you state that the AV cueing task remained unaffected, which does not seem to be the case for the right targets in the post phase of the spatial training.

Line 568 “in our experimental training there was no time restriction on sound duration, and participants experienced the acoustic inputs if they preferred (approximately 4 seconds).”

Do you know how long the participants took to answer (hence to listen to the stimuli) and could you analyse that?

In relation to your remark about the sound duration in line 566 "because the sounds lasted only 1.6 seconds" => 1.6 sec is quite a long duration. Goossens and Van Opstal (1999) observed heading with stimuli lasting only 500 ms.

Line 574-579 “First, reaching to sound could enhance the spatial coding of sound source location because results from the coordination of different effectors (eyes, head, hand) into a common reference frame (25,45). Second, reaching could help directing attention toward the position occupied by the sound source, making the task more engaging (fundamental to foster relearning, see Dehaene, 2020)(46). Third, reaching to sound could foster active listening through head exploration, as suggested by the documented shared reference frame between reaching motor command and head-orienting (47)."

I do not grasp the difference between the first and third proposition, especially in reference to the work of Boyer et al 2013 (ref 47). Please clarify.

In Supplementary material: The CE is now called “spatial advantage”, a notion that has not been introduced before.

6. PLOS authors have the option to publish the peer review history of their article (what does this mean?). If published, this will include your full peer review and any attached files.

Reviewer #1: No

Reviewer #2: **Yes: **Isabelle Viaud-Delmon

---

## [Author Response · Author response to Decision Letter 0]

7 Dec 2021

We thank the Editor for allowing us to revise and resubmit our work. We are also grateful to the Reviewers for the thoughtful suggestions on the previous version of the manuscript. We think the manuscript has greatly improved during the revision process. We have attached the file with the detailed Respond in the attachments.

---

## [Decision Letter · Decision Letter 1]

10 Jan 2022

PONE-D-21-25716R1Adapting to altered auditory cues: generalization from manual reaching to head pointingPLOS ONE

Dear Dr. Valzolgher,

Thank you for submitting your manuscript to PLOS ONE. After careful consideration, we feel that it has merit but does not fully meet PLOS ONE’s publication criteria as it currently stands. Therefore, we invite you to submit a revised version of the manuscript that addresses the points raised during the review process.

We look forward to receiving your revised manuscript.

Kind regards,

Welber Marinovic

Academic Editor

PLOS ONE

Journal Requirements:

Additional Editor Comments:

The reviewers agree that the most concerns have been handled adequately. Reviewer 1 had a few minor requests that I believe the authors will be able to tackle quickly. I also believe I can assess the minor corrections myself, without help from the reviewers, and come to a final decision quickly.

Reviewers' comments:

Reviewer's Responses to Questions

**Comments to the Author**

1. If the authors have adequately addressed your comments raised in a previous round of review and you feel that this manuscript is now acceptable for publication, you may indicate that here to bypass the “Comments to the Author” section, enter your conflict of interest statement in the “Confidential to Editor” section, and submit your "Accept" recommendation.

Reviewer #1: All comments have been addressed

Reviewer #2: All comments have been addressed

2. Is the manuscript technically sound, and do the data support the conclusions?

Reviewer #1: Yes

Reviewer #2: Yes

3. Has the statistical analysis been performed appropriately and rigorously? 

Reviewer #1: Yes

Reviewer #2: Yes

4. Have the authors made all data underlying the findings in their manuscript fully available?

Reviewer #1: Yes

Reviewer #2: Yes

5. Is the manuscript presented in an intelligible fashion and written in standard English?

Reviewer #1: Yes

Reviewer #2: Yes

6. Review Comments to the Author

Reviewer #1: The revised manuscript is much improved. I have just a few minor comments and corrections.

The pictorial legend in Figure 4a remains reversed. Spatial – non spatial training should be open symbols.

Line 421. I could not find Figure S2 in the Supplementary materials. Should it be Table S2?

Table S1. There was a significant interaction between Testing Phase and Azimuth for absolute errors. Does this mean that both spatial and non-spatial training were effective in reducing error on the plugged side?

Table 1. Non-spatial training was also effective in reducing errors, albeit to a lesser extent than spatial training. This is addressed in the paragraph starting line 601, but it would be useful to include this in the Results.

Table S2 columns headers are -10 and 10. Should they be -15 and 5 given the offset between the controller and the speaker?

Line 602. Training was not effective in improving vertical localisation. It could be the case that the reduced range of elevation (20 degrees) made improvements difficult to see, compared to the larger range of azimuths. Monaural cues to elevation are likely to be quite similar over a small range.

Line 670. “overall” should read “overlap”

As suggested by the authors in the response to reviewers, it would be useful to include the data in the figures as well as the models to improve transparency of the results.

There remain a few errors of English expression, primarily in the use of singular/plural.

Reviewer #2: (No Response)

7. PLOS authors have the option to publish the peer review history of their article (what does this mean?). If published, this will include your full peer review and any attached files.

Reviewer #1: No

Reviewer #2: **Yes: **Isabelle Viaud-Delmon

---

## [Author Response · Author response to Decision Letter 1]

20 Jan 2022

We added the Response to Reviewers as file.

---

## [Editor Report · Decision Letter 2]

21 Jan 2022

Adapting to altered auditory cues: generalization from manual reaching to head pointing

PONE-D-21-25716R2

Dear Dr. Valzolgher,

We’re pleased to inform you that your manuscript has been judged scientifically suitable for publication and will be formally accepted for publication once it meets all outstanding technical requirements.

Kind regards,

Welber Marinovic

Academic Editor

PLOS ONE

---

## [Editor Report · Acceptance letter]

2 Apr 2022

PONE-D-21-25716R2 

Adapting to altered auditory cues: generalization from manual reaching to head pointing 

Dear Dr. Valzolgher:

I'm pleased to inform you that your manuscript has been deemed suitable for publication in PLOS ONE. Congratulations! Your manuscript is now with our production department. 

Kind regards, 

on behalf of

Dr. Welber Marinovic 

Academic Editor

PLOS ONE